# T cell differentiation drives the negative selection of pathogenic mitochondrial DNA variants

Imogen G Franklin[1],*, Paul Milne[2],* ⓘ, Jordan Childs[1], Róisín M Boggan[1], Isabel Barrow[1,3,4], Conor Lawless[1], Gráinne S Gorman[1,3,4], Yi Shiau Ng[1,3], Matthew Collin[2],†, Oliver M Russell[1,4],†, Sarah J Pickett[1],† ⓘ

Pathogenic mitochondrial DNA (mtDNA) single-nucleotide variants are a common cause of adult mitochondrial disease. Levels of some variants decrease with age in blood. Given differing division rates, longevity, and energetic requirements within haematopoietic lineages, we hypothesised that cell-type–specific metabolic requirements drive this decline. We coupled cell-sorting with mtDNA sequencing to investigate mtDNA variant levels within progenitor, myeloid, and lymphoid lineages from 26 individuals harbouring one of two pathogenic mtDNA variants (m.3243A>G and m.8344A>G). For both variants, cells of the T cell lineage show an enhanced decline. High-throughput single-cell analysis revealed that decline is driven by increasing proportions of cells that have cleared the variant, following a hierarchy that follows the current orthodoxy of T cell differentiation and maturation. Furthermore, patients with pathogenic mtDNA variants have a lower proportion of T cells than controls, indicating a key role for mitochondrial function in T cell homeostasis. This work identifies the ability of T cell subtypes to selectively purify their mitochondrial genomes, and identifies pathogenic mtDNA variants as a new means to track blood cell differentiation status.

## Introduction

Mitochondrial diseases, characterised by impaired oxidative phosphorylation (OXPHOS), can be caused by pathogenic variants in either the mitochondrial or nuclear genomes. The most common cause of mitochondrial disease in adults is the mitochondrial DNA (mtDNA) m.3243A>G variant ([1]), which is located in the *MT-TL1* gene and primarily causes complex I (NADH: ubiquinone oxidoreductase) dysfunction via disruption of translation ([2]). Carriers presented a range of clinical phenotypes from asymptomatic to a severe neurological condition known as MELAS (mitochondrial encephalomyopathy, lactic acidosis, and stroke-like episodes) syndrome. As mtDNA exists in multiple copies per cell, mixed WT and variant populations can co-exist, giving rise to heteroplasmy, which varies between individuals, tissues, and cells; therefore, the relationship between tissue variant levels and both disease severity and phenotypic presentation is complex ([3], [4], [5], [6], [7]).

Unlike most of the heteroplasmic mtDNA point mutations, there is strong evidence for negative selection against m.3243A>G mtDNA molecules within mitotic tissues, particularly in the blood ([3], [4], [6], [8], [9], [10], [11]). Blood represents an ideal tissue to study selection pressure as it is composed of multiple different cell types within the same microenvironment and is easily accessible. Previous modelling results are consistent with a decline initiating in haemopoietic stem cells, with a mechanism involving the random segregation of mtDNA molecules at mitosis followed by the elimination of cells that have reached a biochemical threshold because of the accumulation of high levels of variant mtDNA ([10]). More recently, an enhanced decline of m.3243A>G within T cells has been demonstrated ([12]); this is consistent with a proposed mtDNA bottleneck within lymphoid cell development ([13]). However, this study did not analyse progenitor cells, and so the question of how progress through the lymphoid differentiation bottleneck drives m.3243A>G decline remains.

Given differing division rates, longevity, and energetic requirements within both the myeloid and lymphoid compartments, we hypothesised that the mutation level would be determined by development stage within each haematopoietic lineage. To address this, we coupled FACS, which allowed a finer resolution of cellular phenotype, with a novel, high-throughput single-cell mtDNA-sequencing technique to investigate m.3243A>G mutation load within 13 blood cell compartments. We compared this with a second variant, m.8344A>G, which, to date, has not been reported to decline with age ([14]). We show that mtDNA mutation level tracks with T cell differentiation status, with less mature, naïve cells exhibiting higher mutation levels.

[1]Wellcome Centre for Mitochondrial Research, Translational and Clinical Research Institute, Newcastle University, Newcastle upon Tyne, England    [2]Haematopoiesis and Immunogenomics Laboratory, Translational and Clinical Research Institute, Newcastle University, Newcastle upon Tyne, England    [3]NHS Highly Specialised Service for Rare Mitochondrial Disorders, Newcastle upon Tyne NHS Foundation Trust, Newcastle upon Tyne, England    [4]NIHR Newcastle Biomedical Research Centre, Newcastle upon Tyne Hospitals NHS Foundation Trust and Newcastle University, Newcastle upon Tyne, England

Correspondence: sarah.pickett@ncl.ac.uk; matthew.collin@ncl.ac.uk
*Imogen G Franklin and Paul Milne contributed equally to this work
†Matthew Collin, Oliver M Russell, and Sarah J Pickett contributed equally to this work

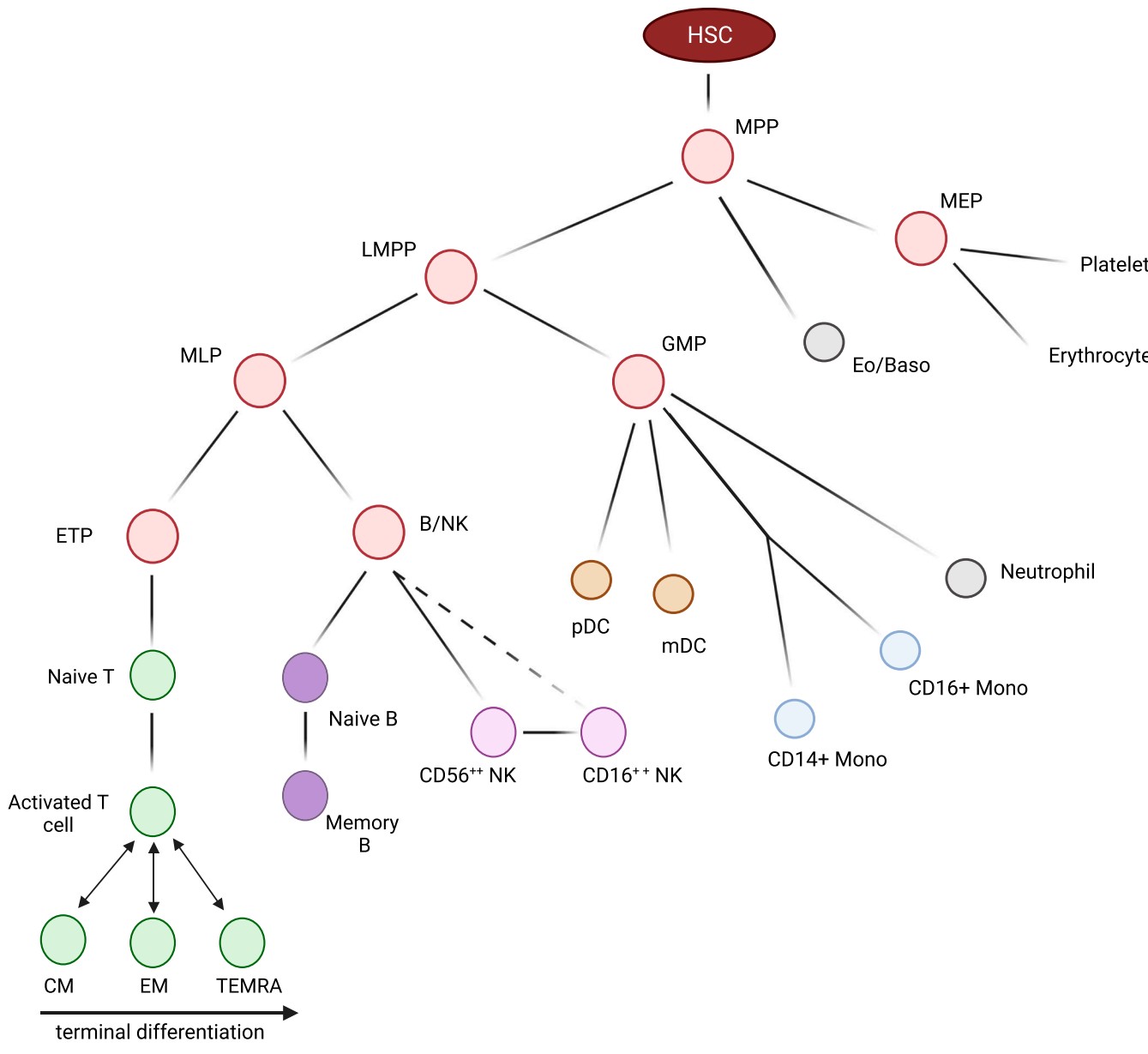

**Figure 1. Investigated blood cell types.**
Haematopoietic differentiation: blood cells are derived from haematopoietic stem cells in the bone marrow; these give rise to highly proliferative CD34+ progenitors (red) which occur infrequently in peripheral blood. Monocyte (blue), granulocytes (grey), dendritic (orange), and natural killer (NK; light purple) cells are involved in the innate immune response. B (dark purple) and T (green) lymphocytes communicate with these cells to generate the highly specific adaptive immune response; B and T memory cells remain after clearance of a pathogen to defend against reinfection. Dashed line represents differentiation yet to be validated. HSC, haematopoietic stem cell; MPP, multipotent progenitors; LMPP, lymphoid-primed multipotent progenitors; MLP, multilymphoid progenitors; ETP, early T cell precursor; B/NK, B and NK cell progenitor GMP, granulocyte–macrophage progenitors; MEP, megakaryocyte–erythroid progenitor; Eo/Baso, eosinophil/basophil; CM, central memory T cell; EM, effector memory T cell; TEMRA, terminal effector cell (CD45RA+ effector memory T cell); NK, natural killer cell; pDC, plasmacytoid dendritic cell; mDC, myeloid dendritic cell.

# Results

### Mitochondrial disease patients have a lower proportion of T cells

Blood samples from 22 patients genetically confirmed to harbour m.3243A>G and 16 controls were fractionated by density centrifugation followed by FACS to separate whole blood into a broad range of cell types of differing maturities (n = 13; Figs 1 and S1). We first investigated whether the presence of m.3243A>G affected cellular proportions. We found that individuals carrying m.3243A>G have significantly lower proportions of T cells (P = 0.0001) and a higher proportion of CD3− HLA-DR+ antigen-presenting cells and B cells (P = 0.0001 and P = 0.0045, respectively) than controls (Fig 2A). Within the T cell compartment, patients and controls have similar CD4+: CD8+ and naïve:memory ratios (Fig 2Bi). There were also no differences in naïve:memory ratios for B cells and CD56 bright/dim for

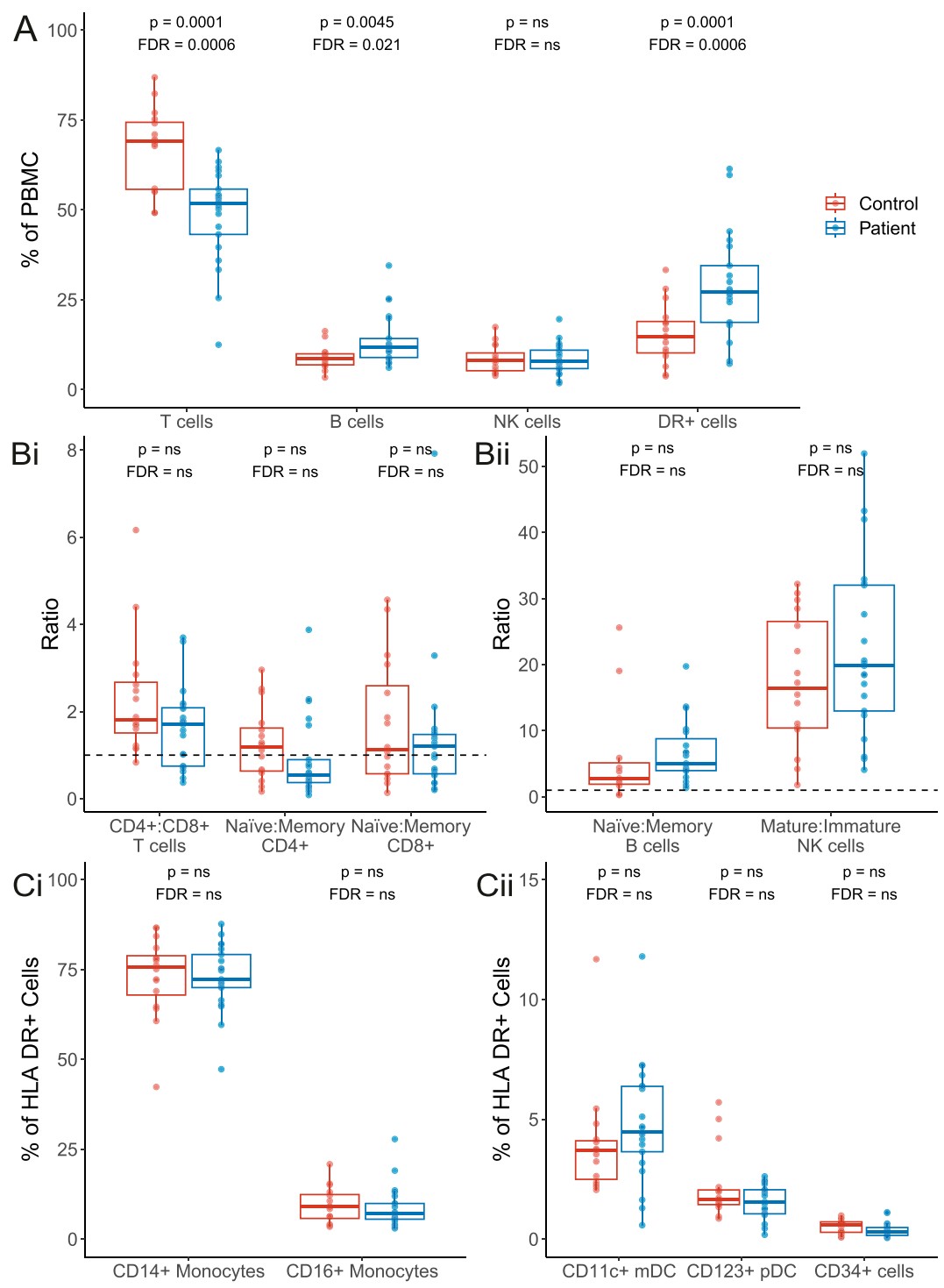

**Figure 2. Comparison of the proportions of cellular subsets in patients and controls.**
Percentage quantification for each cellular subset was derived from flow cytometry data for 21 patients (blue) and 16 controls (red); data were unavailable for P12. See the Materials and Methods section for gating strategy. Values in patients and controls were compared using linear regression, controlling for age at sample; both *P*-values (upper) and values adjusted for (false discovery rate; lower) for all comparisons in the figure are shown; "ns" denotes values >0.05. **(A)** Proportion of T, B, NK, and HLA DR[+] cells. **(B)** Ratio of CD4[+]:CD8[+] T cells, and naïve:memory CD4[+] and CD8[+] T cells, B cells, and NK cells; a ratio of 1 is represented with a dotted line. **(C)** Proportion of subsets of HLA CD[+] cells. DR[+] cells: HLA DR[+] cells CD34[+]: precursor cells; NK, natural killer cells; CD11c[+] mDCs: CD11c[+] myeloid dendritic cells; CD123[+] pDCs: CD123[+] plasmacytoid dendritic cells; PBMC, peripheral blood mononuclear cells.

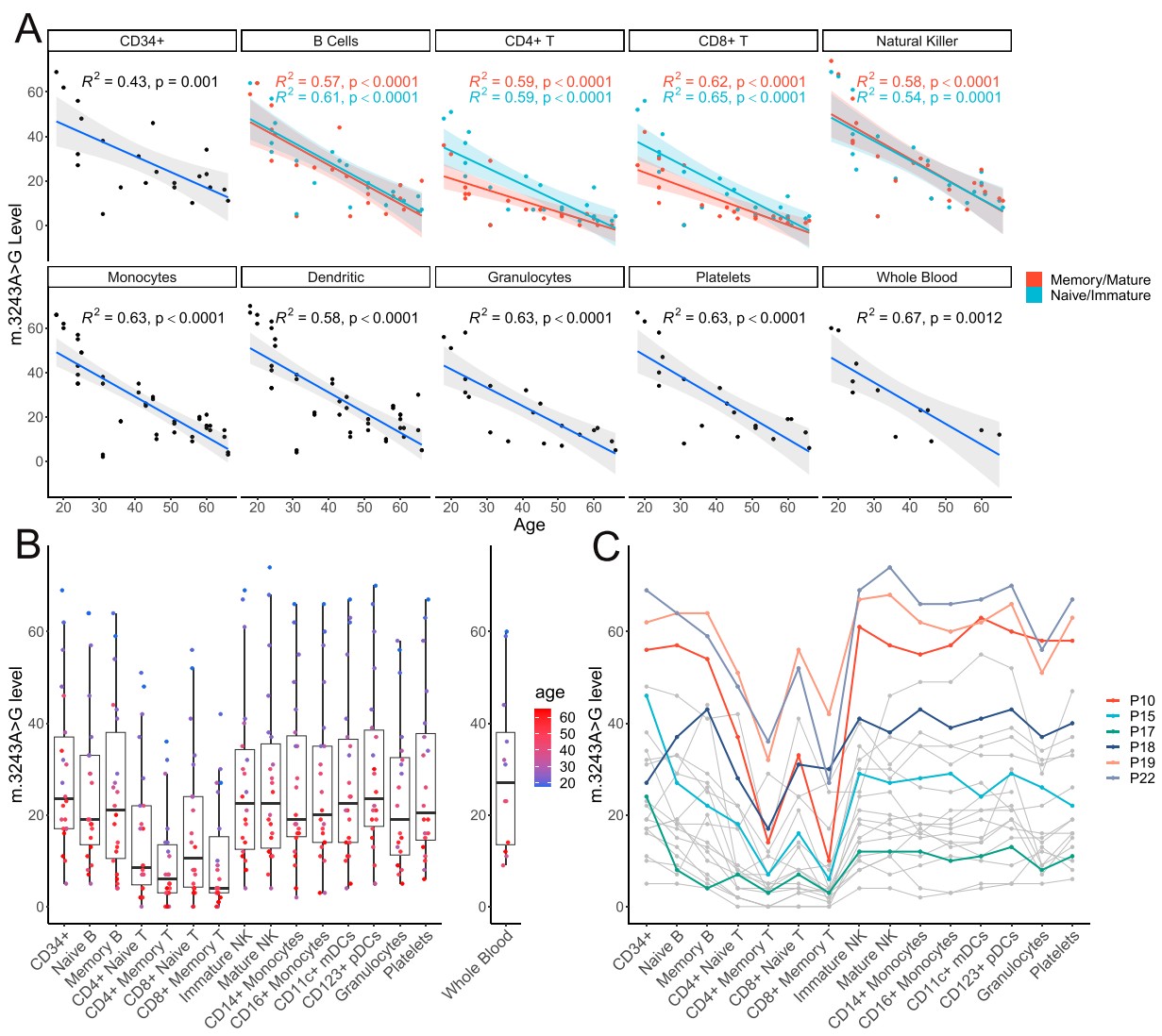

**Figure 3. m.3243A>G levels across bulk populations of peripheral immune cells.**

Thirteen blood cell populations were separated by flow cytometry (n = 22); levels in platelets (n = 20) and granulocytes (n = 20) separated using density centrifugation from the same whole-blood sample are also shown. Each point in the flow cytometry-derived cell populations represents m.3243A>G levels in ~1,000 cells isolated from a single individual. **(A)** m.3243A>G level is negatively correlated with age in all cell populations studied. Naïve/immature cell populations are represented in blue and memory/mature populations in red. Lines represent linear models with 95% confidence intervals; adjusted $R^2$ and $P$-values are shown. Linear mixed models of the effect of age and maturity on m.3243A>G levels including a term for the interaction of age with maturity show that m.3243A>G levels are lower in memory versus naïve CD4[+] and CD8[+] T cells ($P = 0.0034$, $P = 0.0011$) but not B cells ($P = 0.7370$) or mature versus immature NK cells ($P = 0.5275$). The slope is also significantly less steep for naïve CD4[+] and CD8[+] T cells ($P = 0.0411$ and $P = 0.0223$), but not for B or NK cells ($P = 0.9898$ and $P = 0.6146$). **(B)** All cell types studied within the T cell compartment have significantly reduced m.3243>G levels compared with CD34[+] precursor cells, the most naïve blood cells investigated (linear mixed model accounting for patient as a fixed effect; CD4[+] memory: $\beta = -19.55$, SE = 1.66, $P < 0.0001$; CD8[+] memory: $\beta = -18.82$, SE = 1.666, $P < 0.0001$; CD4[+] naïve: $\beta = 12.82$, SE = 1.66, $P < 0.0001$; CD8[+] naïve: $\beta = -12.18$, SE = 1.66, $P < 0.0001$) and the memory B cells ($\beta = -4.36$, SE = 1.66, $P = 0.0090$) and granulocytes ($\beta = -5.72$, SE = 1.70, $P = 0.0009$). **(C)** Where available, m.3243A>G levels in whole blood are also shown (n = 12) (C) The same data are represented with lines connecting points from individual patients. Colours represent samples from the six patients taken forward for single-cell analysis. CD34[+]: precursor cells; immature NK: immature natural killer cells; mature NK: mature natural killer cells; CD11c[+]: myeloid dendritic cells; CD123[+] pDCs: CD123[+] plasmacytoid dendritic cells.

NK cells (Fig 2Bii). Monocyte and dendritic cell proportions do not differ between patients and controls (Fig 2C).

## Selection against m.3243A>G is strongest within the memory T cell compartment

To understand the cell-type–specific relationship of m.3243A>G levels with age, we measured m.3243A>G levels within these blood cell populations from 22 patients. All cellular populations studied showed a significant age-associated decline in m.3243A>G levels ($P$-value range = <0.0001–0.0012; Fig 3A).

Comparison of m.3243A>G levels in all cells relative to CD34[+] haematopoietic progenitors revealed an enhanced clearance of the m.3243A>G variant allele in all T cell subsets (reduction range = 12.18–19.55%, $P$-values < 0.0001). This is consistent between both CD4[+] and CD8[+] T cells. Interestingly, naïve T cells had significantly

higher m.3243A>G levels than their memory cell counterparts (Fig 3A; CD4+: *P* = 0.0034; CD8+: *P* = 0.0011), indicating that mutation levels further reduce as T cells proliferate and differentiate into memory cells. We did not see this maturity-associated enhanced decline in B or NK cell lineages (*P* values > 0.05).

For both CD4+ and CD8+ T cells, negative correlations between age and mutation levels are higher in naïve cells compared with their memory counterparts (Fig 3A; interaction *P* = 0.0411 and *P* = 0.0223), consistent with a weaker negative section at lower m.3243A>G levels. Interestingly, for one patient, there was no detectable m.3243A>G present in any of the T cell populations studied (Fig 3C; P05, whole

blood m.3243A>G level 6%), implying that although negative selection may be weaker, it is still present at low m.3243A>G levels.

We also detected a small reduction in mutation levels in memory B cells and granulocytes (Fig 3B; reduction range = 4.36–5.72%; P = 0.0090 and P = 0.0009, respectively) compared with CD34+ progenitor cells. For the granulocytes, which were isolated by density centrifugation rather than FACS, we cannot rule out a small degree of contamination with other cell types including T cells which are the most abundant type of mononuclear cell. Unlike T cells, there was no significant difference between the levels in naïve and memory B cells.

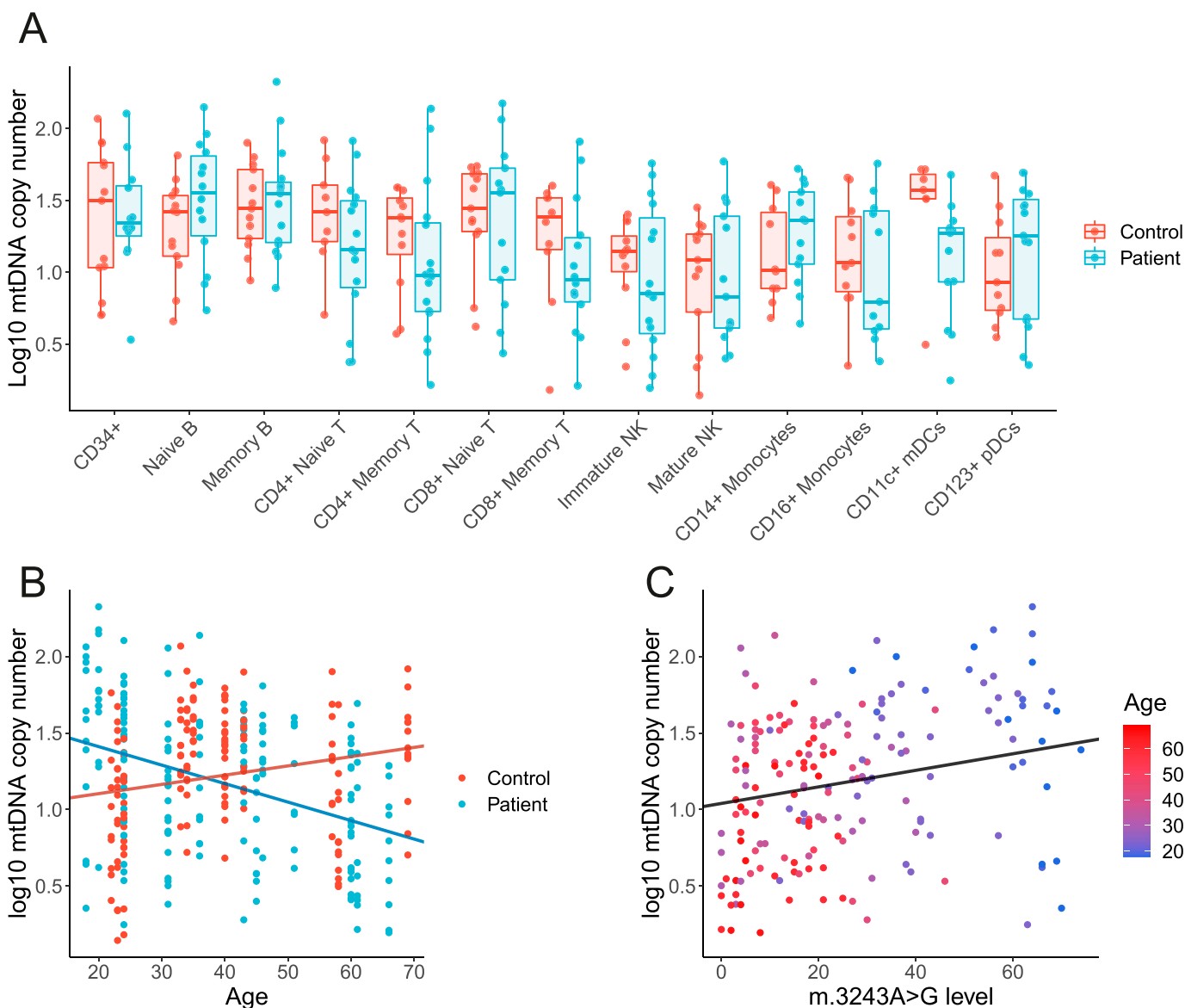

**Figure 4. Mitochondrial DNA (mtDNA) copy number in bulk populations of peripheral immune cells.**
Each point represents relative per cell mtDNA copy number derived from ~1,000 cells from a population of cells, separated using flow cytometry, from a single individual. **(A)** Comparison of log mtDNA copy number between patients (blue; n = 16) and controls (red; n = 14). mtDNA CN is higher in controls than patients in CD11c+ mDCs (Wilcoxon test; *P* = 0.0186, false discovery rate adjusted = 0.2418) but not in other subsets (*P* > 0.05). **(B)** Age and log mtDNA copy number are negatively correlated in m.3243A>G patients (linear mixed model; *β* = −0.0121, SE = 0.0031, *P* = 0.0017) but not in controls (*β* = 0.0060, SE = 0.0046, *P* = 0.2133). **(C)** m.3243A>G level and log mtDNA copy number are weakly positively correlated (linear mixed model; *β* = 0.0054, SE = 0.0024, *P* = 0.0263). Points are coloured by age; as m.3243A>G level in the blood declines with age, the effect of age is difficult to distinguish from m.3243A>G level.

## mtDNA copy number decreases with age in mitochondrial disease patients

m.3243A>G is functionally recessive; low levels of WT mtDNA are capable of rescuing defective oxidative phosphorylation in vitro (15, 16) and higher mtDNA copy number (CN) in skeletal muscle is associated with a lower disease burden (6). Therefore, we sought to investigate the relationship between mtDNA CN and m.3243A>G in blood cells.

For most cell types studied, mtDNA CN is similar between m.3243A>G patients and controls (Fig 4A; Wilcoxon tests; $P > 0.05$), with the exception of CD11c+ myeloid DCs ($P = 0.0186$), although this was not significant after correcting for multiple testing (false discovery rate adjusted = 0.2418). Interestingly, mtDNA CN is negatively correlated with age in cells from m.3243A>G patients (Fig 4B; linear mixed model; $β = -0.0121$, SE = 0.0031, $P = 0.0017$) but there is no relationship in controls ($P = 0.2133$). Given the relationship between age and the m.3243A>G level, this is likely to reflect a positive correlation between m.3243A>G level and mtDNA CN (Fig 4C), suggesting that cells may respond to higher m.3243A>G levels by increasing mitochondrial turnover and perhaps biogenesis. Analysis stratified by cell type reveals that this is likely to be driven by cells of lymphoid origin; mtDNA CN is negatively associated with age and positively associated with the m.3243A>G level in patient cells of lymphoid origin (Fig S2; CD4[+] T cells, CD8[+] T cells, B cells, and NK cells). In T cells, the relationship with the mutation level is higher ($β$ ranges; T cells: 0.0229–0.0234; B and NK cells: 0.0096–0.0106), indicating that T cells are more sensitive to the negative effects of m.3243A>G than other blood cell types. Although there are no overall differences between mtDNA CN between patients and controls in different cell types, it is interesting to note that in younger individuals (<30 yr), patients have a higher mtDNA CN than controls in CD8[+] T cells, NK cells, and B cells. This effect is reversed in older patients who have a lower mtDNA CN than controls (Table S1 and Fig S2).

## Selection against pathogenic mtDNA variants in the T cell compartment is not limited to m.3243A>G

To understand whether this enhanced selection within the T cell compartment is specific to m.3243A>G, we carried out similar investigations in four individuals carrying the m.8344A>G variant, which is located within *MT-TK* (encoding tRNA[lys]) and has been previously reported to maintain stable levels in the blood with age (14, 17, 18).

Consistent with our findings from m.3243A>G, levels of m.8344A>G are significantly reduced in memory T cells (CD4[+] and CD8[+]; $P < 0.0001$), memory B cells ($P = 0.0038$), and granulocytes ($P = 0.0400$) when compared with CD34[+] progenitors (Fig 5). Although we observe the same trend within naïve T cells, this is not significant.

## Single-cell analysis reveals cells that have reverted to WT

Having identified an enhanced reduction in m.3243A>G levels within T cell populations, and a further reduction particularly within the memory T compartment, we sought to understand the dynamics of this by studying m.3243A>G levels within single cells from six patients (age range = 18–46, whole blood m.3243A>G level range = 9–60%). To achieve this, we devised a novel, high-throughput, targeted next-generation sequencing assay, allowing us to determine the m.3243A>G level within 5,732 single cells which had been sorted by FACS into each specific cell population (Fig S3). To gain insight into this enhanced negative selection within the T cell compartment, we further subdivided memory T cells into effector memory (EM), central memory (CM), and TEMRA cells (effector memory T cells re-expressing CD45RA) (gating strategy, Fig S1).

For most cell populations investigated, the range of single-cell m.3243A>G levels spans almost the entire possible range (Fig 6; range = 0.00–99.23%). This is more obvious in cells from younger patients but is also evident in most of the cell types for P15 (age = 45 yr). Despite a much lower whole-blood mutation level for P17 (12% at 41 yr), four of the nine cell subsets for P17 (age = 46 yr) have cells with >65% m.3243A>G. These single-cell data show that the enhanced negative selection against m.3243A>G in T cells can be explained by an increased number of cells with near-zero levels of mutation, with a trend towards total clearance, rather than a general reduction in levels across all cells. This implies that selection against the G allele continues to occur even at very low mutation levels.

## Enhanced negative selection in all subsets of memory T cells

To determine the proportion of cells that have cleared m.3243A>G within each cell population, we quantified the proportion of cells with a near-zero level of m.3243A>G using Bayesian inference (Fig 7 and Table S2). CD34[+] progenitors and monocytes had similar proportions of cells that had cleared m.3243A>G and this proportion increased with age; a pattern that was seen for all lineages (Table S2). Within the T cell compartment, the proportion increases with maturation; for both CD4[+] and CD8[+] cells, all memory cells had a significantly higher proportion of near-zero cells than the naïve cells in most of the individuals. Therefore, selection against m.3243A>G continues during the transition from naïve to memory T cell.

Results comparing CM and EM cells were varied; CD4[+] EM and CM cells showed a similar proportion of near-zero cells, whereas CD8[+] EM and CM cells were more variable. CD8[+] EM had a larger proportion than CD8[+] CMs in 2/6 individuals (one of which was significantly different), but in 3/6 individuals, the proportions were similar or showed the opposite trend (Fig 7 and Table S2). Interestingly, in 5/6 individuals, CD8[+] TEMRA cells had a lower proportion of cells that had cleared the variant allele than EMs; this difference was significant in 3/6 individuals. B memory cells also have a higher proportion of cells that have cleared the variant allele than naïve B cells; although this difference with maturity is not as clear or as large as that seen in T cells, the trend is present across all individuals and is significant in 2/6 of individuals. Notably, the most mature B cell, the plasma cell, is not found in peripheral blood.

# Discussion

Selection against the m.3243A>G variant in the blood tissue is widely recognised in the literature but the mechanism by which this occurs

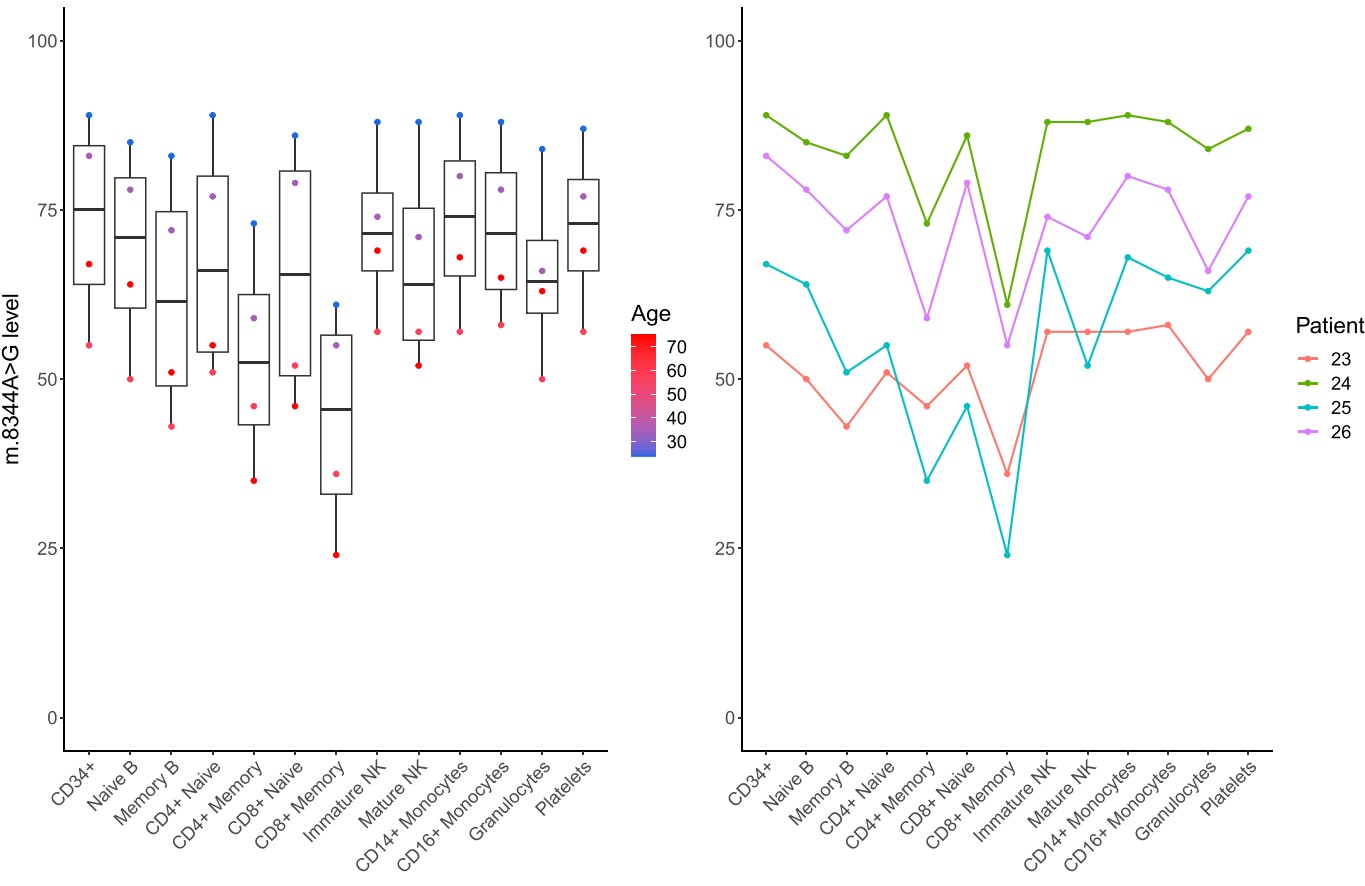

**Figure 5.    m.8344A>G levels across bulk populations of peripheral immune cells.**
Eleven blood cell populations were separated by flow cytometry (n = 4); levels in platelets and granulocytes (n = 4) separated using density centrifugation from the same whole-blood sample are also shown. Each point in the flow cytometry-derived cell populations represents m.3243A>G levels in ~1,000 cells isolated from a single individual as detailed in Fig 1. Memory lymphoid cells and granulocytes possess a m.8344A>G-level significantly reduced from the CD34$^+$ precursor cells (linear mixed model accounting for patient as a fixed effect; CD4 memory: $\beta$ = −20.25, SE = 3.63, $P$ < 0.0001; CD8 memory T: $\beta$ = −29.5, SE = 3.63, $P$ < 0.0001; memory B cells: $\beta$ = −11.25, SE = 3.63, $P$ = 0.0038; granulocytes: $\beta$ = −7.75, SE = 3.63, $P$ = 0.0400).

is unknown (3, 6, 8, 9, 10, 11). Our analysis reveals that this phenomenon is not limited to m.3243A>G, occurs in cells of both lymphoid and myeloid origin, and circulating CD34$^+$ progenitor cells (the earliest cell in haematopoietic differentiation that was investigated), and is particularly enhanced within the T cell compartment. These results are consistent with the research by Walker and colleagues who, via ATAC-seq, demonstrated a significantly increased proportion of lymphocytes with low m.3243A>G levels (12). By using FACs, to achieve a finer resolution of differentiation trajectories, coupled with a customised high-throughput single-cell mtDNA sequencing assay, we show that this phenomenon is enhanced within central and effector memory T cells, and TEMRA cells, demonstrating that the proportion of cells with negligible heteroplasmy continues to increase throughout the T cell lifespan. The trend we observe in the proportion of cells that cleared the mutation supports the conventional view of T cell differentiation, although we were surprised to find that TEMRA cells had a lower proportion of cells that had cleared the variant than EMs, implying either that TEMRA cells are less sensitive to m.3243A>G or that they are not all directly derived from EM cells, according to alternative models of T cell development (19).

Our data also show, for the first time, a reduction in the relative abundance of T cells in m.3243A>G carriers compared with controls, suggesting that the presence of this variant specifically impacts T cell homeostasis, the lineage in which the greatest selective pressure was observed. The difference in T cell abundance remained significant after removal of data from P1, the only patient who had an abnormal lymphocyte count (Table S3). The higher proportion of CD3$^-$ HLA-DR+ cells we observed could simply reflect the lower proportion of T cells, as these are the two largest compartments within PBMC. The link between mitochondrial disease and the immune system is not fully understood, but recurrent infections have been reported in some patients with mitochondrial disease and can result in a severe inflammatory response (20). Interestingly, we did not see a difference in the proportion of T cells in m.8344A>G patients compared with controls; a larger m.8344A>G cohort is needed to elucidate whether this phenomenon is limited to m.3243A>G or also affects patients with other pathogenic mtDNA variants.

T cell development involves a rigorous central tolerance process within the thymus, which involves rapid cell proliferation followed by death of unreactive/autoreactive cells (21). A further

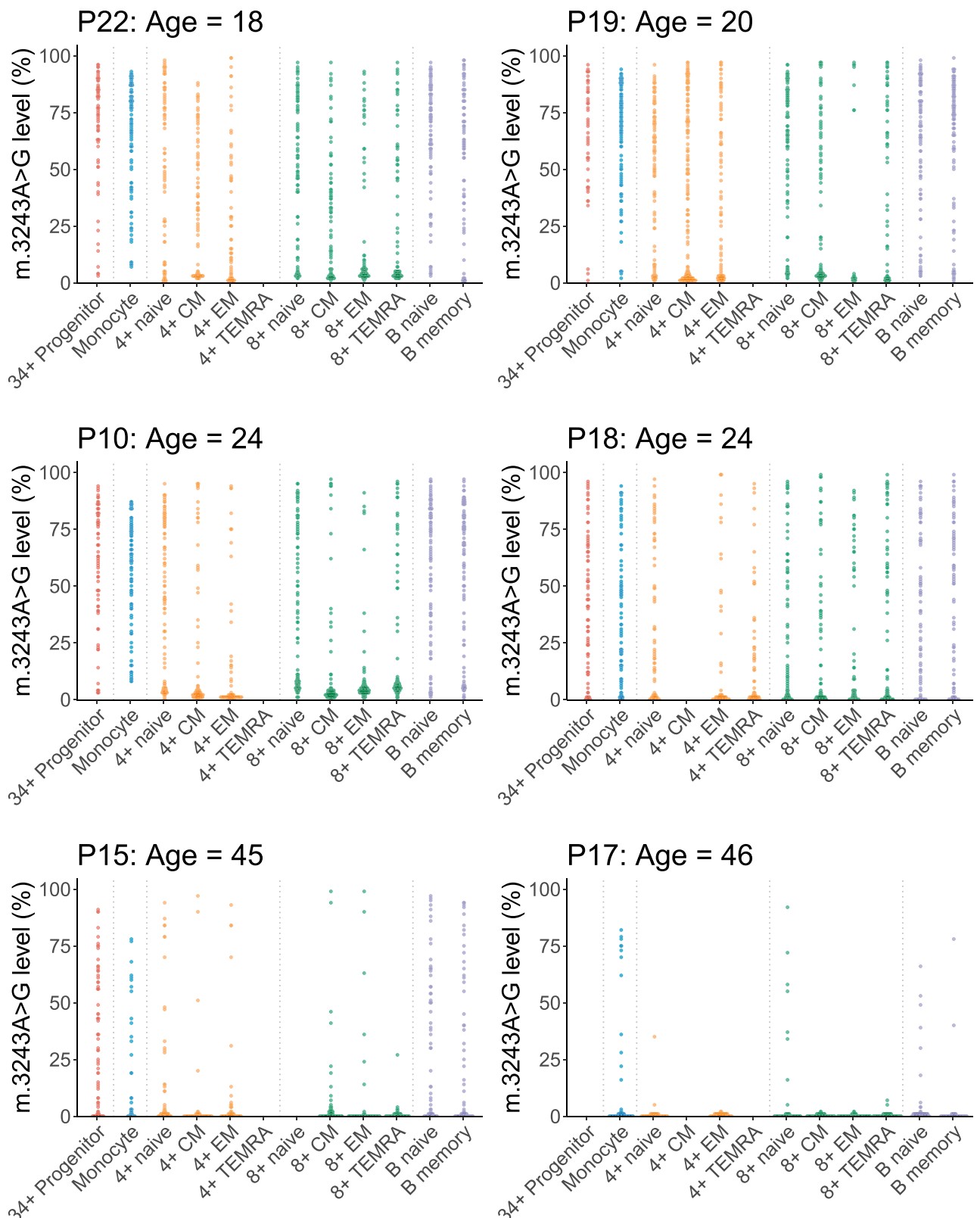

**Figure 6. Dot plots showing single-cell distributions of m.3243A>G levels in peripheral immune cell subsets for six patients.**
Each point represents the m.3243A>G level in a single cell from a single-cell subtype from a single individual (median number of single cells in each subset = 91; IQR = 87, 93, range = 26, 200). Colours represent broad cell subtypes (red: CD34+ precursor cells; blue: monocytes; orange: CD4+ T cells; green: CD8+ T cells; purple: B cells). Where possible, T cells were further separated into naïve, effector memory (EM), central memory (CM), and TEMRA populations and B cells into naïve and memory cells (see Fig S1

proliferative expansion occurs within circulating T cells in response to infection, after which, antigen-experienced memory T cells are poised to mount a response to reinfection (22). Proliferative episodes followed by apoptosis or senescence of cells that have developed a respiratory complex deficiency because of the accumulation of a high mutation load are likely to impact m.3243A>G level distribution within a population (10, 23, 24, 25). However, respiratory complex deficiency can be rescued with as little as 6% WT mtDNA (16), and estimates for the threshold m.3243A>G level in muscle fibres are >80% (26 *Preprint*). It is therefore puzzling that T cells clear the pathogenic variant to levels below the accepted threshold for a biochemical defect for postmitotic tissues. Thus, the mechanism underlying the decline of m.3243A>G, and perhaps other pathogenic mtDNA variants, is either the result of unique metabolic constraints endured by immune cells or not wholly attributable to OXPHOS deficiency at a single-cell level. Statistical modelling of m.3243A>G decline is consistent with selection occurring at the stem-cell level (10); our data demonstrate that levels in CD34$^+$ progenitor cells are negatively correlated with age and so support this theory. The discovery that m.8344A>G shows an enhanced decline in selected lineages, despite relatively stable levels in whole blood (14), indicates that specific immune cells are more sensitive to m.8344A>G, or pass through more rounds of purifying selection than stem cells. This raises the question of whether T cell sensitivity is a general phenomenon among pathogenic mt-tRNA variants and could be related to defects in mitochondrial translation (27). Future work should expand this study to include a wide range of variants within mtDNA protein-coding genes and other mt-tRNA genes.

The drivers of selection against pathogenic mtDNA variants in mitotic tissues are unknown, but our findings suggest that studying this phenomenon in T cells, which show enhanced selection following the hierarchy of differentiation, will be valuable in characterising this process. The relationship between a T cell and its mitochondria is complex; cellular metabolism within the T cell is modified throughout its lifespan and plays a key role in determining cell fate (28, 29). Both proliferating immature thymocytes and activated T cells rely heavily on glucose to regenerate ATP via glycolysis to support rapid proliferation, which might imply that selection against OXPHOS-deficient cells is low at this point. In contrast, circulating naïve and memory T cells, which turnover slowly, are more reliant on OXPHOS (22, 30). Only a subset of activated T cells differentiate into long-lived memory cells; their long-term cellular survival and function are uniquely dependent upon spare respiratory capacity, facilitated by enhanced fatty acid oxidation, and associated with higher expression of complex I (31). Therefore, naïve T cells containing mtDNA mutations are likely to be compromised, perhaps setting the stage for becoming vulnerable to m.3243A>G-induced mitochondrial dysfunction upon activation.

T cell activation is associated with an increased membrane potential (32). Inhibition of OXPHOS complexes I and IV, both of which show deficiency in the presence of m.3243A>G (2, 33, 34, 35, 36, 37), induce activation defects in both CD8$^+$ and CD4$^+$ T cells (32, 38,

39, 40). In addition, inhibiting glycolysis shifts T cell fate away from senescence and towards a self-renewal phenotype (41). Therefore, glycolysis alone is insufficient to sustain T cell activation; functional OXPHOS is vital at this stage of differentiation and may explain why we see the lowest levels of m.3243A>G in cells which have undergone activation-induced mass clonal expansion. This is consistent with observations in m.3243A>G patient fibroblasts where restricting access to glucose and glutamine forces a reliance on pyruvate as a substrate, driving positive selection for WT and negative selection against variant mtDNA molecules (42). This selection is likely to be the result of selective depolarisation of mitochondria containing variant mtDNA, inhibiting the replication of variant mtDNA molecules, and increasing WT mtDNA replication (42). In T cells, aged mitochondria are asymmetrically segregated into daughter cells with higher levels of autophagy, mitochondrial clearance, and self-renewal upon T cell activation-related cell division (41). It is possible that a similar mechanism identifies mitochondria with higher mutation levels and that asymmetric segregation followed by selective mitophagy may contribute to the enhanced reduction in mutation levels that we observe in memory T cells. In our data, we see a higher mtDNA CN in cells of lymphoid origin in younger patients who have higher m.3243A>G levels; this may be related to a higher mitochondrial mass, compensating for a biochemical defect. Interestingly, in older patients, this phenomenon appears to be reversed. Unpicking the processes involved in controlling cellular mtDNA content and their contribution to the decline of m.3243A>G should be a focus of future work in this area.

The mTOR pathway has been shown to be chronically activated in m.3243A>G patient fibroblasts; inhibition of this pathway with rapamycin reduces fibroblast m.3243A>G levels (43). The mTOR pathway is also linked to activation-induced T cell mitochondrial biogenesis (44) and leucine, acting via this pathway plays a key role in T cell activation (45). This relationship is intriguing given that m.3243A>G disrupts mt-tRNA$^{Leu\ (UUR)}$, and may imply that selection against m.3243A>G is linked to amino acid homeostasis. How this links to selection against m.8433A>G (within mt-tRNA$^{Lys}$) is unclear. Therefore, further investigation into selection against other pathogenic mtDNA variants in mt-tRNA and protein coding genes, and the potential link to the mTOR pathway is merited.

Despite being well characterised in whole blood, genetic heterogeneity of mtDNA mutation level and the complex cellular composition of blood tissue have both previously hampered our understanding of the cellular mechanisms involved in the selection against mtDNA mutations. Here, we demonstrate the power of combining innovative single-cell genomic technologies with high-resolution cell phenotyping to gain further insight into the processes underlying blood cell mutation decline and the relationship between T cell development and mitochondrial function. Future studies into the strikingly high levels of negative selection in T cells, taking advantage of the rapid development of omic technologies that can interrogate cellular pathways at the single cell-level, will give us insights into the key processes involved in this mechanism. This has the potential to identify

for gating strategy). Sufficient CD4$^+$ TEMRA cells were only present in one patient (P18). Data for some cellular subsets were removed during data quality control and are shown as missing data (P18: CD4$^+$ CM; P15: CD8$^+$ naïve; P17: CD34$^+$ precursors, CD4$^+$ CM; see the Materials and Methods section for quality-control details).

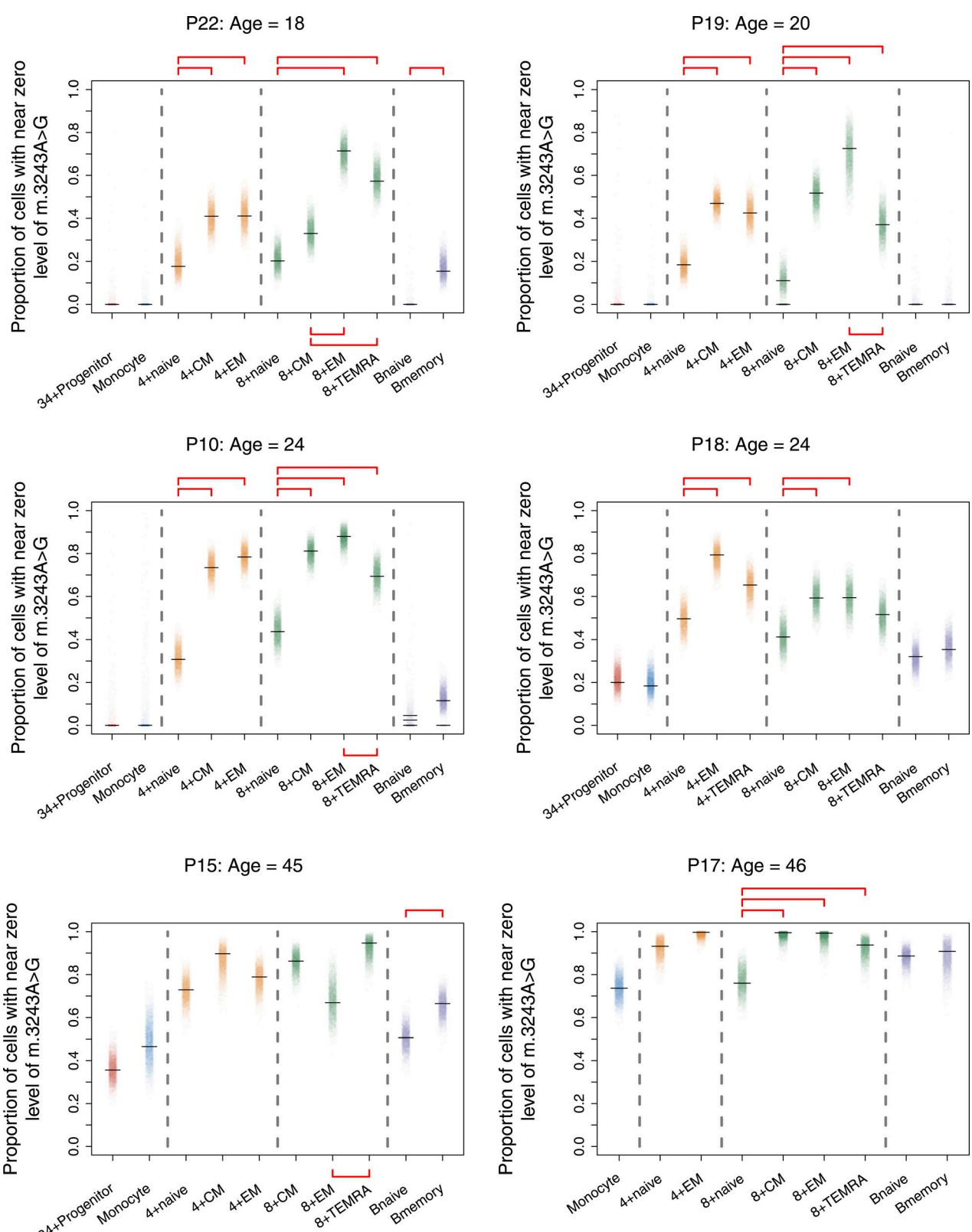

**Figure 7. Proportion of cells that have cleared m.3243A>G increases with cell maturity.**
Each point represents the estimate from one Bayesian iteration. Peak estimates are indicated with horizontal black lines; multiple lines are shown when the posterior distribution is bi/multimodal. Significance testing was carried out between cell-type pairs within major cell-type groups (indicated with horizontal dotted lines); red

drug targets that could be useful in the treatment of mito-chondrial disease, and to inform our understanding of the key role that mitochondria play in the differentiation of T cell lineages.

# Materials and Methods

### Participant recruitment

22 individuals (Table S4) carrying the m.3243A>G variant and four carrying m.8344A>G were recruited from the NHS Highly Specialised Service for Rare Mitochondrial Disorders in Newcastle upon Tyne, UK; seven healthy controls via the Dendritic Cell Homeostasis in Health and Disease study (08/H0906/72) and an additional 12 healthy control samples were obtained from the MAGMA study (17/NE/0015) (Table S5). All m.3243A>G patients underwent clinical assessment using the Newcastle Mitochondrial Disease Adult Scale (NMDAS) (46) to evaluate their disease phenotype and severity. None of these patients reported having any infective symptoms or taking any antimicrobial treatment in the preceding 3 wk. This was confirmed by patients' blood cell counts, which were all within normal ranges, apart from P1 who had a mildly reduced lymphocyte count (full blood-count data presented in Table S3). Of the 22 m.3243A>G patients, we identified six, representing a range of ages (18–46 yr) and whole blood m.3243A>G level (9–60%) to take forward to single-cell analysis.

### Sample preparation

Whole-blood samples (8–20 ml, collected in EDTA) were separated via Ficoll density gradient using Lymphoprep (Stem Cell Technologies) into three populations: PBMCs, platelets, and granulocytes. Platelets were harvested from the plasma fraction and a granulocyte pellet was obtained by lysis of the red-cell pellet using 20 ml red-cell lysis buffer (155 mM ammonium chloride, 12 mM sodium bicarbonate, and 0.1 mM EDTA). Blood samples >15 ml were prepared in two aliquots. Granulocyte and platelet populations were pelleted and frozen at −80°C. PBMCs were washed twice with PBS, split into 3–8 1 ml aliquots with freezing media (FBS + 10% DMSO), and stored as viable cells at −80°.

### Cell sorting and lysis

PBMC samples were stained with fluorescently labelled antibodies (Table S6) and sorted with BD ARIA Fusion with a 70-$\mu$m nozzle according to the gating strategy indicated (Fig S1).

#### *Bulk cell sorting*
Up to 1,000 cells of each cell type were sorted into Eppendorf tubes containing 25 $\mu$l lysis buffer (500 mM Tris–HCl, 1% Tween20, dH$_2$O, 20 mg/ml Proteinase K). mtDNA copy number (CN) calculations were adjusted accordingly for rare populations in which fewer

cells were available (e.g., CD34$^+$ progenitor cells). Sorted samples were centrifuged to ensure all cells were submerged in the buffer, lysed by incubating at 55°C for 2 h and 10 min at 95°C, and frozen at −20°C.

#### *Single-cell sorting*
Single cells were sorted into 15 $\mu$l of the lysis buffer (as above) in a 96-well plate. Each plate contained cells of a single-sort gate and included a negative control well.

### Bulk m.3243A>G measurements

Patient m.3243A>G levels were determined via pyrosequencing using the Pyromark Q24 system as previously described and validated (47). Five control samples with known levels (15%, 46%, 70%, 0%, and 0%) allowed for validation of each pyrosequencing run. If any of the control heteroplasmy readings differed from the known m.3243A>G level by more than ±3%, the experiment was repeated. Primers used to target m.3243A>G in the initial DNA amplification reaction were obtained from IDT (sequences according to GenBank accession number NC_012920.1: 50 biotinylated forward: m.3143-3163; reverse: m.3331-3353), and within the pyrosequencing reaction, a reverse sequencing primer was used: m.3244-3258.

### mtDNA copy number

mtDNA copy number was quantified in triplicate using real-time PCR, as previously described (6). Briefly, each 25 $\mu$l of the population lysate was diluted 1:5 with nuclease-free water and 5 $\mu$l of each diluted sample was used per 15 $\mu$l of the reaction. Primer and probe sequences were relative to GenBank number NC_012920.1: forward: m.3485-3504; reverse: m.3553-3532; VIC-labelled MT-ND1 probe (m.3506-3529). Probes contained a nonfluorescent quencher and 3′ MGB moiety. Each 96-well plate contained a standard curve derived from six 10-fold serial dilutions of a plasmid containing one copy of the ND1 target. Plasmid copies per $\mu$l for each dilution were determined using the DNA concentration of the plasmid sample and its molecular weight. R$^2$ values were >0.9992 and gradients fell between −3.264 and −3.449 for all standard curves. Mean threshold cycle (Ct), standard curves, and the number of cells per sample (as counted by the cell sorter) were used to determine the absolute mtDNA copy number per cell for each sample. Within sample outliers and samples with Ct SD > 0.3 or mean Ct > 30 were excluded.

A control DNA sample was included on each plate and used to standardise mtDNA copy number to control for inter-plate variation using the following formula: normalised mtDNA copy number = (absolute mtDNA copy number/on-plate control mtDNA copy number) × mean control mtDNA copy number.

### Single-cell genomics

After cell lysis, 2 $\mu$l of a unique barcoded forward primer (10 $\mu$M; Table S7) was added to each well followed by the addition of 18 $\mu$l

---

horizontal bars indicate pairs of estimates that are considered statistically different (i.e., zero lies outside the 95% credible interval of the posterior difference between the proportions; see the Materials and Methods section for details).

containing high-fidelity Platinum SuperFi II DNA Polymerase and buffer (Thermo Fisher Scientific), 10 mM dNTP, 10 $\mu$M reverse primer (5′ GGTTGGCCATGGGTATGTTG-3′), and nuclease-free dH$_2$O. Samples were then amplified via PCR according to the following cycling conditions: 30 s at 98°C; 35 cycles of 7 s at 98°C, 10 s at 60°C, 30 s at 72°C; 300 s at 72°C.

PCR products from each plate were pooled (10 $\mu$l per well) and unique Ion Xpress adaptors (numbers 46–80) were added, according to the Ion Plus Fragment library kit protocol (Life Technologies). 3 $\mu$l product from each pooled plate was then combined to form the final library, which was quantified using an Aligent Technologies Bioanalyser. Ion Torrent Chip preparation was carried out using a final library concentration of 80 pmol/l using an Ion Chef according to the manufacturer's instructions and the chip was loaded onto an Ion S5 sequencing machine within 15 min of reaction completion.

### Bioinformatics

Per plate BAM files were converted into fastq format using SAMtools v. 1.12 (48). Reads containing well-specific barcode sequences in both the forward and reverse directions were extracted using Sabre (49); these were then realigned to a reference sequence (derived from RefSeq: NC_012920.1) using BWA v.0.7.17 (50), producing SAM files, which were converted to well-specific pileup files using SAMtools v. 1.12 (48). Variants were called using the mpileup2cns function in Varscan v2.3.9 (51) with a minimum average quality of 28, a minimum variant allele frequency of 0.0001, and a minimum coverage of one. Format fields from the vcf file were extracted using VCFtools v.0.1.16 (52) and imported into R v.3.6.0 (53) for downstream quality control and analysis.

Distributions of per-cell read depth per sequencing batch were inspected to determine an optimum lower threshold; cells with a read depth below this were excluded. To avoid the inclusion of wells containing multiple cells, we also excluded cells with a read depth greater than 1.5 times the upper boundary of the interquartile range. Three plates (P15: 8 + Naïve; P17: 34 + Progenitor; and P18: 4 + CM) were excluded because of suspected contamination in the negative controls, and 5,732 cells passed quality control; median read depth was 1,146 (IQR = 903, 2,792, range = 201, 6,975).

### Validation of technique

Amplicons which were not pooled for sequencing underwent a second PCR reaction using primers detailed in *Bulk heteroplasmy measurements*. This product was then sequenced using pyrosequencing and the m.3243A>G level measured. Comparison of the IonTorrent mutation level estimate and the pyrosequencing level estimate revealed the two techniques to be highly comparable (mean difference = 2.86%, median = 2.95%; with an overall difference in the range of 0.07–6.87%) (Fig S4).

### Statistics

All statistical tests were performed in R v.3.6.0 (53). Linear mixed models were fitted using the nlme package (54) and included random intercepts for ID to account for variation between individuals. All graphs were produced using the ggplot2 package (55).

We considered *P*-values < 0.05 to be significant and, where appropriate, *P*-values were adjusted for multiple comparisons

using the p.adjust function in R using the method = "false discovery rate" option. Comparisons between two groups were performed using the Wilcoxon test. Where shown, box-plots depict the median, and 25th and 75th quantiles and whiskers extend from the hinge to the largest/smallest values no further than 1.5 × IQR from the hinge.

To determine the proportion of cells with near-zero levels of m3243A>G, a Bayesian mixture model was fitted to the single-cell mutation load data for each set of lineage observations for each subject. The model identified the spike in the data (near-zero mutation load) and the proportion of cells which belonged to it. A mixture prior was placed on the proportion of cells belonging to the spike, imposing a probability of a near-zero mutation load.

The model is described as $Y \sim \pi N_0^1(\mu, \sigma^2) + (1 - \pi)U(0, 1)$, $\pi \sim \pi_0 \delta_0 + (1 - \pi_0)U(0, 1)$.

Where $N_0^1$ denotes a normal distribution truncated on range (0,1), that is, only values within this range can have a non-zero density and $\delta_0$ denotes the Dirac delta function—a function whose value is zero everywhere except at zero, implying that $\pi 0$ is the probability of no spike. The priors placed on the unknown parameters were $\mu \sim U(0, 0.2)$, $\sigma \sim Exp(5)$ and $\pi_0 \sim U(0, 1)$. Inference was carried out using the package rjags v.4.13 (56).

### Study approval

Written informed consent from participants was obtained before participation. All clinical investigations were evaluated according to the Declaration of Helsinki. Ethical approval was granted by the Newcastle and North Tyneside Research Ethics Committee (REC:19/LO/0117; "Understanding the decline in levels of the mtDNA mutation m.3243A>G within blood cell subtypes"). Additional control samples were obtained from the "Dendritic Cell Homeostasis in Health and Disease study" (REC:08/H0906/72) and the "MAGMA study" (REC:17/NE/0015).

## Data Availability

Underlying data and supporting analytic code for the manuscript can be accessed via https://github.com/sarahjpickett/mtDNA_variant_blood_cells.

## Supplementary Information

## Acknowledgements

We would like to thank all the individuals who agreed to participate in this study and the clinical, laboratory, and research administration and support teams. We would also like to thank Helen Tuppen for help with optimising the IonTorrent single-cell assay; Doug Turnbull and John Grady for helpful discussions leading up to this study; Rob Taylor and the Newcastle NHS Highly Specialised Laboratory team for assistance with pyrosequencing assays; Andrew Schaefer, Catherine Feeney, and Rhys Thomas for their role in recruiting and clinically assessing patients; Laura Brown and Clare Massarella for assistance with the ethical approval for this study; and Doug

Jerry for clinical data management. This research made use of the Rocket High Performance Computing service at Newcastle University. We acknowledge the Newcastle University Flow Cytometry Core Facility (FCCF) for assistance with the generation of Flow Cytometry data. This work was supported by a Wellcome Career Re-entry Fellowship to SJ Pickett (204709/Z/16/Z); a MRC DiMeN DTP studentship to IG Franklin, the Wellcome Centre for Mitochondrial Research (203105/Z/16/Z) and the UK NHS England Specialist Commissioners which funds the Highly Specialised Service for "Rare Mitochondrial Disorders of Adults and Children" in Newcastle upon Tyne. M Collin and P Milne are supported by Wellcome Trust Investigator Award 219562/Z/19Z. Funding for open-access charge: Charity Open Access Fund (COAF). For the purpose of open access, the author has applied a CC BY public copyright licence to any author-accepted manuscript version arising from this submission. Figs 1 and S3 and the graphical abstract were created with BioRender.com.

## Author Contributions

IG Franklin: data curation, formal analysis, investigation, and writing—original draft, review, and editing.
P Milne: investigation, methodology, and writing—review and editing.
J Childs: formal analysis.
RM Boggan: investigation.
I Barrow: investigation.
C Lawless: formal analysis.
GS Gorman: resources.
YS Ng: resources, investigation, and methodology.
M Collin: conceptualization, resources, supervision, funding acquisition, and writing—original draft, review, and editing.
OM Russell: conceptualization, resources, supervision, funding acquisition, and writing—original draft, review, and editing.
SJ Pickett: conceptualization, resources, data curation, formal analysis, supervision, funding acquisition, and writing—original draft, review, and editing.

## Conflict of Interest Statement

The authors declare that they have no conflict of interest.

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
