## [Reviewer comments · Life Science Alliance]

Life Science Alliance

T cell differentiation drives the negative selection of pathogenic mitochondrial DNA variants

Imogen Franklin, Paul Milne, Jordan Childs, Roisin Boggan, Isobel Barrow, Conor Lawless, Grainne Gorman, Yi Ng, Matthew Collin, Oliver Russell, and Sarah Pickett

DOI: <https://doi.org/10.26508/lsa.202302271>

Corresponding author(s): Sarah Pickett, Newcastle University

Review Timeline:

Submission Date:	2023-07-13
Editorial Decision:	2023-07-13
Revision Received:	2023-07-18
Editorial Decision:	2023-07-24
Revision Received:	2023-08-08
Accepted:	2023-08-09

Transaction Report:

Please note that the manuscript was previously reviewed at another journal and the reports were taken into account in the decision-making process at *Life Science Alliance*.

Reviews

Referee #1 Review

Remarks for Author:

Summary

In the manuscript by Franklin et al., the authors investigate the basis for purifying selection against the human m.3243 A<G pathogenic variant in the haematopoietic compartment. It is well established in the field that there is negative selection against this variant in the blood of patients, but the mechanisms remain elusive. Here, the authors perform detailed analysis of different haematopoietic cell lineages from patients and controls at different ages coupled with deep sequencing approaches to investigate this outstanding question of mitochondrial genetics. What is abundantly clear from the data is that there is negative selection against the m.3243 A<G pathogenic variant in all cell lineages with age. In fact, it is quite remarkable the consistency in the rate of selection independent of the cell lineage. Where the authors do identify a discrepancy is the rate between naïve and memory T cells. This is seen for both CD4+ and CD8+ lineages. However, this effect is only observed in the younger patients and is not apparent in older individuals from the data presented. If a selection mechanism is operating at an intracellular level, then the differences between memory and naïve T cells could simply be due to the age of the cells per se in the circulation and nothing to do with the T cell differentiation process itself. As a result, the evidence to support the claim in the title of the manuscript is lacking.

Major points

1. The authors do not comment on the dependence of age for the effect between memory and naïve T cells. Memory T cells in younger individuals could simply have had more time for the intracellular selection process to eliminate the m.3243 A<G pathogenic variant compared to naïve T cells. Can the authors demonstrate that the effect is not due simply to difference in T cell age?
2. Did the authors consider isolating the different T cell lineages followed by cell culturing with stimulation to proliferate? In a sense, to test whether the rate of cell proliferation enhance the selection mechanisms against the m.3243 A<G pathogenic variant in those individual T cell lineages.
3. Although putative mechanisms for the selective pressure against the m.3243 A<G pathogenic variant are only raised in the Discussion; the explanations are purely speculative. Especially considering how robust this mtDNA selection mechanisms is across all haematopoietic cell lineages as exemplified by the data from the authors.
4. Although the authors were careful to include only patients and controls who did not succumb to an infection three weeks before the blood sampling, understanding how infections affect the mtDNA selection mechanism against the m.3243 A<G pathogenic variant would be particularly valuable to this study and the field, especially considering the experimental approach established by the authors.

Minor points

1. For the m.8344 A>G pathogenic variant was a similar analysis done as in Figure 2 for the m.3243 A<G variant?

Referee #2 Review

Comments on Novelty/Model System for Author:

Selection against mutant 3243G>A mtDNA is known. Some novelty comes from identifying T-cell population as a major component of the previously observed mutant mtDNA decrease in blood.

The medical applicability of the observations is limited as there is no new therapeutic insight into mutant mtDNA clearance. There are also no new diagnostic or clinical outcome insights for patients. Furthermore, the studied diseases mainly affect muscle and neurones, so relevance to therapeutic alleviation of disease symptoms is also unclear. In the long run, however, these observations add to our understanding of the role of functional OXPHOS in cell proliferation and clearance of mutant mtDNA.

The model system is adequate in studying the fate of pathogenic mtDNA mutations in blood.

Remarks for Author:

In the presented manuscript, Franklin and coworkers have been using flow cytometry combined with single cell sequencing to follow the heteroplasmy levels of two pathogenic mtDNA variants in different blood cell types. The results confirm previous reports of negative selection against the pathogenic variants in blood, as evident from the negative correlation of their levels with age. The novelty of the study is that the group has pinpointed the selection to occur specifically in the T-cell pool. This is quite an interesting finding, which also potentially reveals something about the mutant mtDNA clearance mechanisms in the stem or progenitor cells.

Overall, the paper is well written and appears methodologically sound. I have only few suggestions.

Chapter 3, end of the page: "For most cell types studied, mtDNA CN is similar between m.3243A>G patients and controls [...] Given the relationship between age and m.3243A>G level, this is likely to reflect a positive correlation between m.3243A>G level and mtDNA CN, suggesting that cells respond to higher m.3243A>G levels by increasing mitochondrial biogenesis."

- If this was the case, one would expect the CN to be higher in the patients than in controls? It is an interesting observation, especially considering that there was no correlation between CN and age in the controls. Would it rather reflect a turnover of mitochondria with high heteroplasmy levels, resulting in CN decrease together with the observed elimination of the mutants? Is there any evidence of increased autophagy in the patient T-cell pool? This might be possible to quantify using FACS (see eg. doi: 10.1007/978-1-0716-2071-7_5). Given that mTOR pathway (mentioned in the discussion to be chronically activated in 3243A>G patients) is also controlling autophagy, this might be interesting to check.

The role of OXPHOS in T-cell biology is reviewed in the discussion and is relevant for explaining the findings. However, it is unclear to me what is different in the T-cell biology compared to B- or NK-cells, to explain the sensitivity of the former to OXPHOS deficiency? Also "In our data, we see a higher mtDNA CN in cells of lymphoid origin in patients with higher m.3243A>G levels; this may be related to a higher mitochondrial mass, compensating for a biochemical defect." -> But there was no CN difference between patients and controls, the correlation within patients CN was related to age, where younger patients had higher mutant loads and CN (Figure 4C)? It is possible to measure mitochondrial mass from the cells, if you insist on this hypothesis.

Minor points:

Page numbers are missing.

Space before the reference parentheses is missing throughout the text.

July 13, 2023

Re: Life Science Alliance manuscript #LSA-2023-02271-T

Dr. Sarah Jane Pickett
Newcastle University
Wellcome Trust Centre for Mitochondrial Research
Medical School, Newcastle University
Framlington Place
Newcastle upon Tyne NE2 4HH
UNITED KINGDOM

Dear Dr. Pickett,

Thank you for submitting your manuscript entitled "T cell differentiation drives the negative selection of pathogenic mtDNA variants" to Life Science Alliance. We invite you to submit a revised manuscript addressing the following Reviewer comments:

- Address Reviewer 1's major point #1 and the minor point.
- Address Reviewer 2's points.

Thank you for this interesting contribution to Life Science Alliance. We are looking forward to receiving your revised manuscript.

Sincerely,

- A letter addressing the reviewers' comments point by point.
- An editable version of the final text (.DOC or .DOCX) is needed for copyediting (no PDFs).
- High-resolution figure, supplementary figure and video files uploaded as individual files: See our detailed guidelines for preparing your production-ready images, <https://www.life-science-alliance.org/authors>
- Summary blurb (enter in submission system): A short text summarizing in a single sentence the study (max. 200 characters including spaces). This text is used in conjunction with the titles of papers, hence should be informative and complementary to the title and running title. It should describe the context and significance of the findings for a general readership; it should be written in the present tense and refer to the work in the third person. Author names should not be mentioned.
- By submitting a revision, you attest that you are aware of our payment policies found here: <https://www.life-science-alliance.org/copyright-license-fee>

B. MANUSCRIPT ORGANIZATION AND FORMATTING:

Referee #1 (Remarks for Author):

Summary

In the manuscript by Franklin et al., the authors investigate the basis for purifying selection against the human m.3243 A<G pathogenic variant in the haematopoietic compartment. It is well established in the field that there is negative selection against this variant in the blood of patients, but the mechanisms remain elusive. Here, the authors perform detailed analysis of different haematopoietic cell lineages from patients and controls at different ages coupled with deep sequencing approaches to investigate this outstanding question of mitochondrial genetics. What is abundantly clear from the data is that there is negative selection against the m.3243 A<G pathogenic variant in all cell lineages with age. In fact, it is quite remarkable the consistency in the rate of selection independent of the cell lineage. Where the authors do identify a discrepancy is the rate between naïve and memory T cells. This is seen for both CD4+ and CD8+ lineages. However, this effect is only observed in the younger patients and is not apparent in older individuals from the data presented. If a selection mechanism is operating at an intracellular level, then the differences between memory and naïve T cells could simply be due to the age of the cells per se in the circulation and nothing to do with the T cell differentiation process itself. As a result, the evidence to support the claim in the title of the manuscript is lacking.

We agree that the difference in m.3243A>G level between naïve and memory T cells is greatest in younger individuals (figure 2) – the lack of a difference in older individuals is likely due to levels approaching 0%. Our single cell data corroborate this – in the two patients over 45yrs age, the majority of memory cells have completely cleared the variant.

Major points

1. The authors do not comment on the dependence of age for the effect between memory and naïve T cells. Memory T cells in younger individuals could simply have had more time for the intracellular selection process to eliminate the m.3243 A<G pathogenic variant compared to naïve T cells. Can the authors demonstrate that the effect is not due simply to difference in T cell age?

This comment appears to imply that purifying selection might occur through an alternative but unspecified time-dependent process that separates naïve and memory T cells and is distinct from the process of differentiation. However, 'simply had more time' is not credible as an alternative explanation to explain the differences between naïve and memory cells, in our view. Although the process of differentiation is necessarily time-dependent, it is brief, occurring over a very short interval of weeks (PMID:29236685).

Most importantly, there is no evidence to suggest that there is any appreciable difference between the mean calendar age of naïve and memory T cell within a given individual. Naïve T cells are not younger than memory T cells; their principal difference relates to experience of antigen. C¹⁴ studies show unequivocally that naïve T cells self-renew at a slow rate by peripheral turnover and thus, age progressively with the individual (PMID:31661488). Metabolic labelling estimates put the life-span of both naïve and memory T cells at between 500-1500 days in humans with a longer estimate for naïve T cells (PMID: 32322253). Further evidence of their similarity in age comes from the observation that cord blood has equal proportions of naïve and memory cells, both generated within the preceding 6 months.

In our view, therefore, the most likely explanation for the observation that memory T cells have undergone enhanced purifying selection, compared with naïve T cells, is the process of differentiation that occurs as individual naïve clones transition to memory clones over the life-time of the individual. Differentiation requires many cycles of proliferation over a short interval of time which naturally offers scope for metabolic selection to have occurred.

A small number of memory T cells from P15 (Age 45; Figure 6) have retained high m.3243A>G levels, suggesting that these cells were made in early life. This agrees with a very recent study (published whilst our manuscript was under review) which observed enhanced clearance of large scale pathogenic mtDNA deletions within T cells in paediatric patients, but in adults the only cells in which the deletion could still be detected were memory T cells (PMID: 37386249). The retention of pathogenic variants in a small number of these long-lived, largely quiescent cells suggests that selection is unlikely to be an ongoing intracellular process and is more likely to occur during the expansion phase.

In response to this comment, we have added the following sentence to the discussion acknowledging that selection follows T cell differentiation hierarchy but emphasising that further work is needed to determine when and how this selection is occurring.

“The drivers of selection against pathogenic mtDNA variants in mitotic tissues are unknown, but our findings suggest that studying this phenomenon in T cells, which show enhanced selection following the hierarchy of differentiation, will be valuable in characterising this process.”

2. Did the authors consider isolating the different T cell lineages followed by cell culturing with stimulation to proliferate? In a sense, to test whether the rate of cell proliferation enhance the selection mechanisms against the m.3243 A<G pathogenic variant in those individual T cell lineages.

Yes, we have considered this. However, our experience of culturing mitotic cells that carry m.3243A>G is that they tend to lose the variant upon cell proliferation. Therefore, understanding this process will require experiments that investigate not only the decline of m.3243A>G with proliferation but also the effects of the variant on T cell function and the ability of naïve cells to differentiate into memory cells. These experiments will take a considerable amount of time, would significantly delay the publication of the important findings we report and so we believe they are outside the scope of this manuscript.

3. Although putative mechanisms for the selective pressure against the m.3243 A<G pathogenic variant are only raised in the Discussion; the explanations are purely speculative. Especially considering how robust this mtDNA selection mechanisms is across all haematopoietic cell lineages as exemplified by the data from the authors.

We agree that, at this stage, we can only speculate on the potential mechanisms of decline. However, our findings will act as a foundation for this emerging field, prompting functional studies to characterise the exact mechanisms involved, some of which we have discussed.

4. Although the authors were careful to include only patients and controls who did not succumb to an infection three weeks before the blood sampling, understanding how infections affect the mtDNA selection mechanism against the m.3243 A<G pathogenic variant would be particularly valuable to this study and the field, especially considering the

experimental approach established by the authors.

Yes, the effect of infection / vaccination on selection, as well as the effect of m.3243A>G on immune response will be a fascinating future direction for this project.

Minor points

1. For the m.8344 A>G pathogenic variant was a similar analysis done as in Figure 2 for the m.3243 A<G variant?

Yes, no significant difference was found but we only have data for four m.8344A>G patients and so lack statistical power to detect differences in cell proportions. We have commented on this in the discussion:

“Interestingly, we did not see a difference in the proportion of T-cells in m.8344A>G patients compared to controls; a larger m.8344A>G cohort is needed to elucidate whether this phenomenon is limited to m.3243A>G or also affects patients with other pathogenic mtDNA variants”.

Referee #2 (Comments on Novelty/Model System for Author):

Selection against mutant 3243G>A mtDNA is known. Some novelty comes from identifying T-cell population as a major component of the previously observed mutant mtDNA decrease in blood.

The medical applicability of the observations is limited as there is no new therapeutic insight into mutant mtDNA clearance. There are also no new diagnostic or clinical outcome insights for patients. Furthermore, the studied diseases mainly affect muscle and neurones, so relevance to therapeutic alleviation of disease symptoms is also unclear. In the long run, however, these observations add to our understanding of the role of functional OXPHOS in cell proliferation and clearance of mutant mtDNA.

The model system is adequate in studying the fate of pathogenic mtDNA mutations in blood.

Referee #2 (Remarks for Author):

In the presented manuscript, Franklin and coworkers have been using flow cytometry combined with single cell sequencing to follow the heteroplasmy levels of two pathogenic mtDNA variants in different blood cell types. The results confirm previous reports of negative selection against the pathogenic variants in blood, as evident from the negative correlation of their levels with age. The novelty of the study is that the group has pinpointed the selection to occur specifically in the T-cell pool. This is quite an interesting finding, which also potentially reveals something about the mutant mtDNA clearance mechanisms in the stem or progenitor cells.

Overall, the paper is well written and appears methodologically sound. I have only few suggestions.

1. We would like to thank the reviewer for recognising our interesting findings and the quality of our manuscript.

Chapter 3, end of the page: "For most cell types studied, mtDNA CN is similar between m.3243A>G patients and controls [...] Given the relationship between age and m.3243A>G level, this is likely to reflect a positive correlation between m.3243A>G level and mtDNA CN, suggesting that cells respond to higher m.3243A>G levels by increasing mitochondrial biogenesis."

- If this was the case, one would expect the CN to be higher in the patients than in controls? It is an interesting observation, especially considering that there was no correlation between CN and age in the controls. Would it rather reflect a turnover of mitochondria with high heteroplasmy levels, resulting in CN decrease together with the observed elimination of the mutants? Is there any evidence of increased autophagy in the patient T-cell pool? This might be possible to quantify using FACS (see eg. doi: 10.1007/978-1-0716-2071-7_5). Given that mTOR pathway (mentioned in the discussion to be chronically activated in 3243A>G patients) is also controlling autophagy, this might be interesting to check.

2. We thank the reviewer for highlighting this discrepancy in our interpretation. We agree that if the negative correlation of mtDNA CN with age was entirely due to increased mitochondrial biogenesis, we would expect to see a higher mtDNA CN in patient compared to controls. Upon revisiting the data, we found that for younger individuals (<30 years), patients have a higher mtDNA CN than controls in CD8+ T cells, NK cells and B cells. This effect is reversed in older patients who have a lower mtDNA CN than controls, which, as the reviewer pointed out, may reflect increased autophagy or a higher mtDNA turnover rate in patients.

We have modified the sentence quoted by the reviewer above to reflect this and have added these analyses into Table S6:

*"Given the relationship between age and m.3243A>G level, this is likely to reflect a positive correlation between m.3243A>G level and mtDNA CN (Figure 4c), suggesting that cells **may** respond to higher m.3243A>G levels by increasing mitochondrial **turnover and perhaps** biogenesis"*

We have also added the following text to the Results and Discussion to acknowledge this:

Results: “Although there are no overall differences between mtDNA CN between patients and controls in different cell types, it is interesting to note that in younger individuals (<30 years), patients have a higher mtDNA CN than controls in CD8+ T cells, NK cells and B cells. This effect is reversed in older patients who have a lower mtDNA CN than controls (Table S7, Figure S2).”

Discussion: “In our data, we see a higher mtDNA CN in cells of lymphoid origin in **younger** patients, who have higher m.3243A>G levels; this may be related to a higher mitochondrial mass, compensating for a biochemical defect. Interestingly, in older patients, this phenomenon appears to be reversed. Unpicking the processes involved in controlling cellular mtDNA content and their contribution to the decline of m.3243A>G should be a focus of future work in this area.”

Cell Type	Age < 30 years			Age >= 30 years		
	Difference in median mtDNA CN	p value	n	Difference in median mtDNA CN	p value	n
Dendritic	-0.0135	0.9399	22	0.0462	0.7060	50
Monocytes	0.5585	0.1469	22	-0.1921	0.3378	50
CD34+	0.6897	0.1143	11	-0.2630	0.2359	25
CD4+ T	-0.0021	0.5185	22	-0.5195	0.0052	50
CD8+ T	0.8781	0.0097	22	-0.6752	0.0001	50
Natural Killer	1.0279	0.0250	22	-0.5580	0.0106	50
B Cells	0.4764	0.0005	22	-0.0776	0.4079	50

Table S7: Difference between log10 mtDNA copy number in m.3243A>G patients and controls, stratified by age and cell type (Wilcoxon test).

The role of OXPHOS in T-cell biology is reviewed in the discussion and is relevant for explaining the findings. However, it is unclear to me what is different in the T-cell biology compared to B- or NK-cells, to explain the sensitivity of the former to OXPHOS deficiency?

3. We did also find enhanced selection against m.3243A>G in memory B cells. However, given the more pronounced effect in T cells, it makes sense for future work to focus on characterising mechanism of selection within these cells. We have added the following sentence to the discussion to acknowledge this.

“The drivers of selection against pathogenic mtDNA variants in mitotic tissues are unknown, but our findings suggest that studying this phenomenon in T cells, which show enhanced selection following the hierarchy of differentiation, will be valuable in characterising this process.”

Also "In our data, we see a higher mtDNA CN in cells of lymphoid origin in patients with higher m.3243A>G levels; this may be related to a higher mitochondrial mass, compensating for a biochemical defect." -> But there was no CN difference between patients and controls, the correlation within patients CN was related to age, where younger patients had higher mutant loads and CN (Figure 4C)? It is possible to measure mitochondrial mass from the cells, if you insist on this hypothesis.

See response number 2, above.

July 24, 2023

RE: Life Science Alliance Manuscript #LSA-2023-02271-TR

Dr. Sarah Jane Pickett
Newcastle University
Wellcome Trust Centre for Mitochondrial Research
Medical School, Newcastle University
Framlington Place
Newcastle upon Tyne NE2 4HH
United Kingdom

Dear Dr. Pickett,

Thank you for submitting your revised manuscript entitled "T cell differentiation drives the negative selection of pathogenic mitochondrial DNA variants". We would be happy to publish your paper in Life Science Alliance pending final revisions necessary to meet our formatting guidelines.

- please upload all figure files as individual ones, including the supplementary figure files
- please add your supplementary figure and table legends to the main manuscript text after the legends for the main figures
- please remove "the paper explained" section
- <https://github.com/sarahjpickett/SingleCellBloodM3243> does not bring us to an operating page, please update this link for data accession

A. FINAL FILES:

B. MANUSCRIPT ORGANIZATION AND FORMATTING:

Sincerely,

August 9, 2023

RE: Life Science Alliance Manuscript #LSA-2023-02271-TRR

Dr. Sarah Jane Pickett
Newcastle University
Wellcome Trust Centre for Mitochondrial Research
Medical School, Newcastle University
Framlington Place
Newcastle upon Tyne NE2 4HH
United Kingdom

Dear Dr. Pickett,

Thank you for submitting your Research Article entitled "T cell differentiation drives the negative selection of pathogenic mitochondrial DNA variants". It is a pleasure to let you know that your manuscript is now accepted for publication in Life Science Alliance. Congratulations on this interesting work.

DISTRIBUTION OF MATERIALS:

Again, congratulations on a very nice paper. I hope you found the review process to be constructive and are pleased with how the manuscript was handled editorially. We look forward to future exciting submissions from your lab.

Sincerely,
